# Risk of Glaucoma Associated with Components of Metabolic Disease in Taiwan: A Nationwide Population-Based Study

**DOI:** 10.3390/ijerph19010305

**Published:** 2021-12-28

**Authors:** Ya-Wen Chang, Fung-Chang Sung, Ya-Ling Tzeng, Chih-Hsin Mou, Peng-Tai Tien, Cheng-Wen Su, Yu-Kuei Teng

**Affiliations:** 1School of Nursing, China Medical University, Taichung 406040, Taiwan; yawen172@mail.cmu.edu.tw (Y.-W.C.); tyaling@mail.cmu.edu.tw (Y.-L.T.); 2Department of Health Services Administration, China Medical University, Taichung 406040, Taiwan; fcsung1008@yahoo.com; 3Management Office for Health Data, China Medical University Hospital, Taichung 404332, Taiwan; b8507006@gmail.com; 4Department of Food Nutrition and Health Biotechnology, Asia University, Taichung 41354, Taiwan; 5Department of Ophthalmology, China Medical University Hospital, Taichung 404332, Taiwan; miketien913@gmail.com; 6Department of Ophthalmology, Asia University Hospital, Taichung 41354, Taiwan; fashiongo0405@gmail.com

**Keywords:** glaucoma, metabolic disease, diabetes mellitus, hypertension, hyperlipidaemia

## Abstract

Purpose: This retrospective cohort study was conducted to determine the glaucoma risk associated with metabolic disease (MetD) using insurance claims data of Taiwan. Methods: From the database, we identified patients with newly diagnosed hypertension, diabetes and/or hyperlipidemia from the years 2000 to 2002 as the MetD cohort (N = 42,036) and an age-gender-diagnosis-date matched control cohort without MetD with a two-fold sample size than that of the MetD cohort. Both cohorts were followed until the development of glaucoma, death, or withdrawal, until 31 December 2013. The incidence of glaucoma, and the Cox method estimated hazard ratio (HR) of glaucoma were calculated. Results showed that the incidence of glaucoma was two-fold higher in the MetD cohort than in the controls (1.99 versus 0.99 per 1000 person-years), with an adjusted HR of 1.66 (95% CI: 1.50–1.85). The glaucoma incidence was higher in patients with diabetes than those with hypertension and hyperlipidemia (2.38 versus 1.95 and 1.72 per 1000 person-years, respectively). The incidence increased to 5.67 per 1000 person-years in patients with all three comorbidities, with an aHR of 4.95 (95% CI: 2.35–10.40). We also found higher incidence rates of primary open-angle glaucoma and primary angle-closure glaucoma with aHRs of 2.03 and 1.44, respectively. It was concluded that glaucoma risk increased with the number of MetD. Health providers need to monitor patients with MetD to prevent glaucoma.

## 1. Introduction

Glaucoma is the second leading cause of irreversible blindness worldwide [1]. Vision loss from glaucoma not only causes significant negative effects on health-related quality of life, but also exerts an increasing economic burden for both patients and society as the disease progresses [2,3,4]. The lifetime indirect costs due to reduced productivity and reduced employment has been estimated to range from USD $5 billion to USD $7 billion [5]. A 3.54% global prevalence of glaucoma has been reported among individuals aged between 40–80 years. The number of individuals with glaucoma worldwide was estimated to be 76.0 million in 2020 and is estimated to increase to 111.8 million by 2040 [5]. Glaucoma is classified into two main subtypes: primary open-angle glaucoma (POAG) and primary angle-closure glaucoma (PACG) [2]. A systematic review and meta-analysis study indicated the estimated pooled global prevalence of POAG to be 3.05% and that of PACG to be 0.50%. As the prevalence of PACG has been the highest in Asians, particular emphasis was made on the development of methods to identify and treat PACG in Asia [6]. POAG is a singular and most common subtype of glaucoma with an open angle, normal-appearing anterior chamber, and the presence of glaucomatous optic disc change and/or visual field defects [2,7]. The mechanism of PACG involves pupillary block and anterior lens movement, leading to angle crowding and, consequently, intraocular pressure [8]. The risks of POAG and PACG are associated with demographic variables such as gender; age; individual socioeconomic status; and comorbidities such as hypertension, hyperglycaemia, and dyslipidaemia [8,9].

Metabolic diseases (MetD) have become a worldwide public health concern because a cluster of conditions may occur together, including key components of hypertension, hyperglycaemia, and dyslipidaemia with low serum high-density lipoprotein and visceral obesity [10,11]. A recent meta-analysis based on 27 studies with 45,811 participants reported that MetD could affect near 24% of type 1 diabetes patients [12]. Studies have also linked MetD with age-related cataract [13], age-related macular degeneration (AMD) [10], obstructive sleep apnea (OSA) [14], nonalcoholic fatty liver disease [15], depression, anxiety [16,17], and hypothyroidism [18]. Nevertheless, particular attention should be paid to the association between MetD and related eye diseases to prevent blindness.

Studies have indicated that hypertension [7,19] and diabetes mellitus [3] were important risk factors for the development and progression of glaucoma [20,21]. By contrast, a Korean cross-sectional study using survey data found that individuals with obesity had a lower prevalence of POAG than the nonobese population did [22]. The consistency of the association between MetD and glaucoma is thus unclear. Furthermore, a meta-analysis with 15 observation studies found that patients with glaucoma are at a high risk of central retinal vein occlusion (CRVO) with an odds ratio of 6.21 [13]. A case-control analysis using insurance data in Taiwan found the PACG cases had significant relationships with cataract and MetD [23]. Therefore, coexistence of cataract with glaucoma reflects that both eye disorders may share similar risk factors.

Studies have shown components of MetD are associated with POAG [20,22] or PACG [23]. However, to the best of our knowledge, no study has used population-based longitudinal data to explore the association between MetD and the risk of developing glaucoma (both PACG and POAG). Therefore, we used claims data from the National Health Insurance (NHI) of Taiwan to perform a retrospective cohort study to explore the development of glaucoma in individuals with and without diagnosed components of MetD.

## 2. Study Population and Methods

### 2.1. Data Source

The present population-based retrospective cohort study retrieved data from the Longitudinal Health Insurance Database (LHID) 2000, which is one of the databases of Taiwan’s NHI Research Database (NHIRD) of all NHI beneficiaries, covering over 99% of Taiwan’s population. LHID 2000 was created for research purposes by systematically and randomly selecting one million individuals from all populations registered in 2000 to represent the entire population. This database included information on demographic status of the beneficiaries and claims data for medical services provided to them from 1996 to 2013. The disease diagnoses were encoded in accordance with the International Classification of Diseases, Ninth Revision, Clinical Modification (ICD-9-CM). To protect personal privacy, the patients’ personal identifications were encrypted from data released from National Health Research Institutes. Therefore, patient consent is not required to access the NHIRD. A large volume of research has been published based on the NHIRD in peer-reviewed international journals. This study was approved by the Ethical Research Committee at China Medical University and Hospital (CMUH104-REC2-115[CR-2]).

### 2.2. Study Cohorts Selection

Figure 1 shows the process of identifying patients for the retrospective cohort study. We used the diagnostic codes of the ICD-9-CM to identify individuals with any of the three major types of MetD including diabetes (ICD-9-CM 250), hypertension (ICD-9-CM 401–405), and hyperlipidaemia (ICD-9-CM 272) for the potential study cohort of MetD. Patients with glaucoma was identified at the baseline and those with missing information regarding gender or age were excluded. Patients who were newly diagnosed as having any of the three types of MetD between 2000 and 2002 (*N* = 42,036) were included as the study cohort of MetD. The first date of diagnosis of MetD was defined as the index date. For each patient with any MetD, two controls free of any MetD were randomly selected from the LHID2000, frequency-matched by sex, age (in 5-year age bands) and index year as controls of the non-MetD cohort (*N* = 84,072). Both cohorts were tracked from baseline until the development of glaucoma, loss to follow-up, withdrawal from NHI, or December 31, 2013 to examine the risk of glaucoma.

### 2.3. Comorbidities and Drugs

Comorbidities that were considered as potential covariates in the inference of the association between MetD and glaucoma included hypothyroidism, OSA, depression, anxiety, headaches, liver diseases, peptic ulcers, cataract, central retinal vein occlusion (CRVO), and age-related macular degeneration (AMD).

Studies have indicated that some medications commonly used in patients may affect both PACG and POAG, including adrenergic drugs, anticholinergics, cholinergics, and sulfa-based medications [24,25]. We also included these drugs in one category as a covariate to evaluate the associated glaucoma risk.

### 2.4. Statistical Analysis

SAS software version 9.1 (SAS Institute, Cary, NC, USA) was used for the data analyses. We compared the distributions of sociodemographic characteristics and baseline comorbidities between cohorts with and without MetD. Medications that might associate with the development of glaucoma were also compared between the two groups. The Chi-square test and Student’s *t* test were used to examine the categorical and continuous data, respectively.

The Kaplan–Meier method was used to calculate and plot the cumulative incidences of glaucoma for both cohorts and the log-rank test was used for comparison. We pooled data of both cohorts to calculate incidence rates of glaucoma (per 1000 person-years) for both cohorts and by sociodemographic status, comorbidity and the use of medication. The hazard ratio (HR) of glaucoma with a 95% confidence interval (CI) was calculated using the Cox proportional hazards regression analysis. The adjusted hazard ratio (aHR) was estimated using the multivariable regression analysis, after controlling for variables significant in measuring the crude HR. We also assessed the incidence rate and HRs of glaucoma associated with individual component and multi-components of MetD. Data analysis further estimated the incidence rate and HRs of PACG and POAG. All analyses were performed with a significance level using α = 0.05.

## 3. Results

### 3.1. Basic Characteristics of Study Cohorts

With a mean age of near 51 years, both cohorts were similar in distributions of sex and age consisting of 54.3% men and 45.7% women, with 59.9% individuals aged 40–64 years old. Near 59% of persons had mean monthly incomes of less than NT$ 20,000 (58.8%). The MetD cohort had significantly higher comorbidities of obesity, hypothyroidism, OSA, depression, headaches, liver diseases, peptic ulcers, cataracts, AMD, and more users of glaucoma associated drug than the controls did (*p* < 0.05) (Table 1). However, prevalence rates of obesity, hypothyroidism, OSA, anxiety, CRVO were less than one percent.

With minimum and maximum follow-up times of 0.003 and near 14.0 years, respectively, the mean follow-up was longer in the MetD cohort than in the control cohort (11.4 (SD 2.97) versus 7.88 (SD 4.62) years). The numbers of study population at risk reduced to 23752 persons and in the MetD cohort and 21,885 persons in the control cohort. Figure 2 shows that the cumulative incidence of glaucoma was 1.5% greater in the MetD cohort than in the controls over the follow-up period (*p* < 0.001).

### 3.2. Association between Metabolic Disease and Glaucoma

Table 2 presents factors associated with the risk of development of glaucoma. The incidence rate of glaucoma was two-fold greater in the MetD cohort than in the control cohort (1.99 versus 0.99 per 1000 person-years) with an aHR of 1.66 (95% CI: 1.50–1.85) (*p* < 0.0001). The incidence of glaucoma was higher in women than in men with an aHR of 1.13 (95% CI = 1.03–1.25). Glaucoma incidence increased with age. Compared with the incidence of patients less than 40 years old, the incidence in those aged 40–64 years was near three-fold higher and that in the older patients was six-fold higher (0.47, 1.48 and 2.82 per 1000 person-years, respectively). Among comorbidities, AMD had the strongest association with the risk of glaucoma (aHR = 2.11; 95% CI: 1.59–2.80), followed by cataract (HR = 1.90, 95% CI = 1.66–2.19) and peptic ulcer (aHR = 1.13; 95% CI: 1.01–1.26). Moreover, the study results indicated that some drugs might have an association with developing glaucoma, but not significantly so (aHR = 1.10; 95% CI = 0.99–1.22).

Among the three types of MetD, individuals with diabetes mellitus had a stronger association with glaucoma than those with other two diseases, with an aHR of 2.14 (95% CI = 1.80–2.55) compared to controls (Table 3). The glaucoma incidence increased with the number of MetD, to 5.67 per 1000 person-years for those with all three diseases with an aHR of 4.95 (95% CI: 2.35–10.4; *p* < 0.001). Among patients having two diseases, the glaucoma incidence was greater in those with diabetes mellitus and hypertension than in those with diabetes mellitus and hyperlipidemia, and those with hypertension and hyperlipidemia (3.77 versus 2.70 and 1.48 per 1000 person-years).

Table 4 shows that the incidence of glaucoma was slightly higher for POAG than for PACG, with aHRs of 2.03 (CI = 1.75–2.36) and 1.70 (CI = 1.53–1.88), respectively, compared with controls.

## 4. Discussion

Through this retrospective cohort study, we found that the incidence of glaucoma (both PACG and POAG) was higher in patients with MetD, which confirms a significant association between MetD and glaucoma. This finding is consistent with findings in previous studies and supports the hypothesis that patients with MetD are at a higher risk of glaucoma than those without MetD are [20]. The global prevalence of POAG was 3.05% and that of PACG was 0.5%. However, the study by Tham et al. indicated that the prevalence of PACG was the highest in the Asian population (2014). Asian patients constitute 76.7% of PACG cases and 53.4% of POAG cases worldwide. Asia may contribute approximately 60% of the glaucoma cases in the world and will still contain the greatest number of patients with POAG and PACG in 2040 [6]. This finding provides evidence of the importance of identifying POAG and PACG for Asian populations.

In our study, we observed that women were more likely than men to develop glaucoma. According to Healthy Vision 2010, women have a higher risk of visual impairment from glaucoma, particularly from PACG [26,27]. The most likely reason for the increased risk of glaucoma for women is that the life expectancy of women is approximately 6 years longer than that of men. Other reasons, such as differences in the propensity to seek medical care or gender inequality in access to medical care, may also contribute to this difference [26,28]. Some biological explanations for this difference have been speculated. First, closed glaucoma in women could be due to the shorter eyes and shallower anterior chamber in women, resulting in limited space at the corner of the eye and impaired outflow of aqueous humor [28]. Secondly, age-related decline in female sex hormones may increase the risk of developing glaucoma. However, insufficient evidence exists regarding the use of hormone replacement therapy to prevent glaucoma [9,28]. Hence, clinicians should pay attention to the development of glaucoma in female patients with MetD.

This study shows that the incidence of glaucoma among the age groups seems to be linear, and is the highest in older patients. Studies have concluded that age is a crucial factor related to the development of glaucoma and that the risk of developing glaucoma increases exponentially with age [27,29]. In addition, women and older individuals with a higher risk of developing MetD also have a higher risk of developing glaucoma [30,31].

Studies have reported that some drugs might be associated with the development of glaucoma [6,25,32]. For example, alpha-adrenergic agonists are found in a variety of drugs, including mydriatics, which ophthalmologists and optometrists usually use in phenylephrine eye drops to dilate the pupils for routine fundus examinations [6,32]. Mydriatics may cause closure of the iridocorneal angle and pupillary block, inducing PACG via two mechanisms [25]. Our study examined whether the development of glaucoma is associated with the use of adrenergic drugs, anticholinergics, cholinergics, and sulpha-based medications. We failed to observe a significant relationship. However, understanding the risk factors associated with glaucoma will help health providers determine which patients would benefit from screening for monitoring of the disease.

### 4.1. Components of Metabolic Disease and Glaucoma

Through this study, we demonstrated the association between MetD and glaucoma. The findings revealed that patients with three types of MetD concurrently, namely diabetes mellitus, hypertension, and hyperlipidaemia are at the highest risk (aHR = 4.95) of developing glaucoma. Individuals with both diabetes mellitus and hypertension also had a higher risk (aHR = 2.87) of developing glaucoma than those with a single MetD did. In addition, we analyzed the relationship between a single MetD and glaucoma. The results indicated that diabetes mellitus was independently associated with a higher risk of the development of glaucoma (HR: 2.14). A longitudinal study demonstrated that diabetes mellitus and hypertension, independently or in combination, were associated with an increased risk of developing POAG [20]. The results of a meta-analysis showed that diabetes mellitus was associated with a significantly increased risk of glaucoma, and the risk of glaucoma increased by 5% each year after the diagnosis of diabetes mellitus [2]. Several possible mechanisms have been inferred as underlying the association between diabetes mellitus and the increased risk of diabetic retinopathy and glaucoma. Diabetes mellitus is associated with abnormalities of lipid metabolism that may promote cellular apoptosis and increase oxidative stress—the same mechanism by which retinal ganglion cell loss occurs in glaucoma [33,34]. A previous study reported that the prevalence of retinopathy in patients with diabetes mellitus ranged 17%–29% in 5 years, and it increased to 78%–100% after 15 years [31]. Moreover, vascular dysregulation [35] of protein kinase C [36] also plays a contributory role in both diabetic eye disease and glaucoma.

Evidence revealed that hyperlipidaemia is significantly associated with an increased risk of glaucoma. In our study, hyperlipidaemia was an independent risk factor (HR: 1.79) for glaucoma, which was similar to the findings of another meta-analysis study [37]. However, a longitudinal study indicated that hyperlipidaemia alone was associated with a slightly reduced (5%) risk of POAG [20]. One possible explanation may be that excessive blood lipid levels increase scleral venous pressure and blood viscosity, resulting in a decrease in outflow facilities [37]. Recently, genetic predisposition has been indicated as another important factor associated with hyperlipidaemia and glaucoma. The ATP binding cassette subfamily A member 1 (ABCA1) may mediate lipid export and nascent high density lipoprotein (HDL) biogenesis [38], and caveolin 1 (CAV1) has been proven to be involved in lipid metabolism and its regulation [39]. These genes might be useful to detect the potential risk of glaucoma. In addition, hypertension was an independent risk factor (HR: 1.48) for glaucoma in our study. This result is similar to that of Langman et al.’s study. The common pathogenesis of ciliary and renal tubular epithelium can explain the co-occurrence of glaucoma and systemic hypertension [40].

The findings of the present study showed that MetD had a significant association with hypothyroidism, OSA, depression, headaches, liver diseases, peptic ulcers, cataract, CRVO, and AMD, consistent with the results of another study in Taiwan [23]. However, the study investigated the role of comorbidities for patients with PACG but not with POAG. A previous study also indicated that thyroid functions (hypothyroidism) affect MetD parameters including triglycerides, HDL cholesterol, plasma glucose, and blood pressure [18]. Glaucoma has a long latency period, in which glaucomatous optic nerve damage continues but remains asymptomatic until later stages. Adherence to regular ophthalmological examinations should be emphasized in patients with MetD, especially among those with multiple diseases [21].

MetD has long been considered a risk factor for glaucoma [20,21,41]. However, no study has used a population-based follow-up design to explore the risk of developing glaucoma (both PACG and POAG) in patients with MetD and to evaluate the combined effects of the three types of MetD. The present study fills this gap and reveals that the risk of glaucoma is associated with MetD independently and jointly. Moreover, glaucoma is one of the most prevalent eye diseases in older adults, and the onset of MetD is usually in middle age [6]. Taken together, these data suggest that the components of MetD contribute to glaucoma development and that the joint effect of several components in middle-aged or older adults might worsen the glaucoma.

### 4.2. Strengths and Limitations

The major strengths of this research are as follows. First, we analyzed the relationship between MetD and the risk of developing glaucoma of two subtypes (POAG and PACG) and found the risk was greater for the subtype of POAG than that of PACG. Second, the study cohorts were matched by age, gender, and index year to exclude potential confounding effects of these variables. Third, the large sample size and long follow-up period increased the validity of the study. Fourth, patients were diagnosed as having MetD and glaucoma by physicians rather than based on patient self-reports. The accuracy of diagnosis is increased.

However, the study also has some limitations. First, information on laboratory and anthropometric measurements is not available in the claims data. We were unable to use measurements of body mass index, waist circumference, blood pressure, blood sugar, triglycerides, and high-density lipoprotein (HDL) to define the MetD cohort for this study. We therefore used the healthcare providers’ diagnoses of hypertension, hyperglycaemia and dyslipidaemia available in the claims data to select the study cohorts. Secondly, information regarding the potential confounding effects of family history and personal lifestyle (including physical activity and smoking history) could not be obtained owing to database limitations. Third, patients with MetD and glaucoma were identified based on the physician’s diagnosis rather than through the research assignment.

Further research is underway to validate the role of glaucoma-related obesity among patients with MetD in the development of glaucoma. Additionally, control of other characteristics related to the development of glaucoma, such as alcohol consumption, smoking, waist circumference, body mass index, and physical activity, must be considered.

## 5. Conclusions

The results of our study showed that the risk of developing glaucoma increased with MetD components. Patients with MetD were at a higher risk for POAG than for PACG compared with patients without MetD. As glaucoma is usually relatively asymptomatic, we recommend that primary care physicians provide appropriate screening for the risk of glaucoma and counsel patients with MetD. It is particularly important to detect glaucoma at an early stage in older patients with diabetes and provide treatment for blindness prevention.

## Figures and Tables

**Figure 1 ijerph-19-00305-f001:**
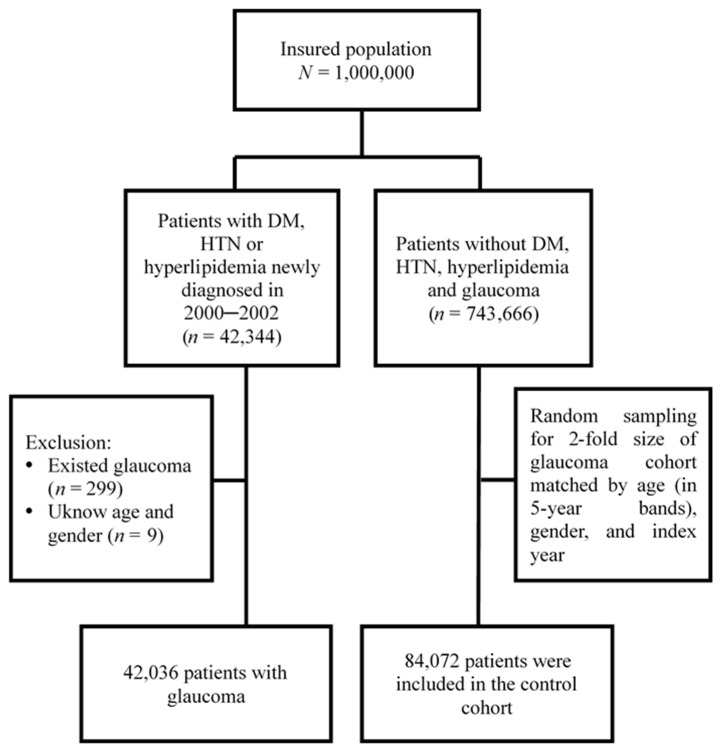
Flow chart of the study population selection.

**Figure 2 ijerph-19-00305-f002:**
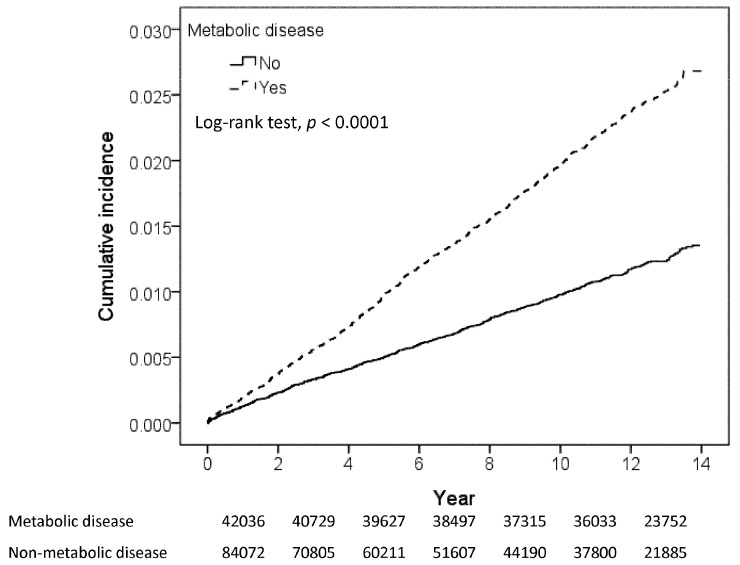
Kaplan-Meier method estimated cumulative incidence of glaucoma.

**Table 1 ijerph-19-00305-t001:** Comparison of demographic characteristics, comorbidities and drug may induce glaucoma between cohorts with and without any metabolic disease (diabetes, hypertension, hyperlipidaemia).

Characteristic	With Metabolic Disease *N* = 42,036	Without Metabolic Disease *N* = 84,072	*p* Value
	*n* (%)	*n* (%)	
Obesity			<0.0001
No	41,762 (99.3)	84,028 (99.95)	
Yes	274 (0.65)	44 (0.05)	
Age, year			0.99
<40	8846 (21.0)	17,692 (21.0)	
40–64	25,194 (59.9)	50,388 (59.9)	
65+	7996 (19.0)	15,992 (19.0)	
Mean(SD)	51.4(14.6)	51.2(14.6)	0.20
Gender			0.99
Female	19,217 (45.7)	38,434 (45.7)	
Male	22,819 (54.3)	45,638 (54.3)	
Monthly income, NTD			0.006
<20,000	24,712 (58.8)	49,579 (59.0)	
20,000–40,000	11,990 (28.5)	24,333 (28.9)	
>40,000	5334 (12.7)	10,160 (12.1)	
Comorbidity			
Hypothyroidism	99 (0.24)	86 (0.10)	<0.0001
OSA	55 (0.13)	30 (0.04)	<0.0001
Depression	1027 (2.44)	1013 (1.20)	<0.0001
Anxiety	49 (0.12)	45 (0.05)	0.0001
Headaches	5670 (13.5)	5843 (6.95)	<0.0001
Liver diseases	9338 (22.2)	7308 (8.69)	<0.0001
Peptic ulcers	12,085 (28.8)	16,543 (19.7)	<0.0001
Cataract	3828 (9.11)	5936 (7.06)	<0.0001
CRVO	9 (0.02)	4 (0.05)	0.014
AMD	473 (1.13)	562 (0.67)	<0.0001
Glaucomas-associated drug	22,356 (53.2)	33,349 (39.7)	<0.0001

NDT: New Taiwan Dollar; OSA: obstructive sleep apnea; CRVO: central retinal vein occlusion; AMD: age-related macular degeneration.

**Table 2 ijerph-19-00305-t002:** Incidence and hazard ratio of glaucoma by metabolic disease, demographic status, comorbidity and drugs may induce glaucoma in pooled study population.

Variables	Glaucoma *n*	Person-Years	Rate	Crude HR (95% CI)	*p*	Adjusted HR (95% CI)	*p*
Metabolic disease							
No	659	662,371	0.99	Ref.		Ref.	
Yes	951	478,927	1.99	2.00 (1.81–2.21)	< 0.0001	1.66 (1.50–1.85)	<0.0001
Obesity							
No	1606	1,137,703	1.41	Ref.			
Yes	4	3595	1.11	0.78 (0.29–2.09)	0.6270		
Age (year)							
<40	136	290,859	0.47	Ref.		Ref.	
40–64	1025	691,465	1.48	3.19 (2.67–3.82)	< 0.0001	2.83 (2.36–3.38)	<0.0001
65+	449	158,975	2.82	6.16 (5.08–7.47)	< 0.0001	3.72 (3.03–4.58)	<0.0001
Gender							
Female	822	522,015	1.57	1.24 (1.12–1.36)	< 0.0001	1.13 (1.03–1.25)	0.0142
Male	788	619,283	1.27	Ref.		Ref.	
Monthly income, NTD							
<20,000	926	656,740	1.41	1.04 (0.89–1.21)	0.6418		
20,000–40,000	479	333,856	1.43	1.06 (0.90–1.24)	0.5206		
>40,000	205	150,702	1.36	Ref.			
Comobidity							
Hypothyroidism							
No	1607	1,139,616	1.41	Ref.			
Yes	3	1682	1.78	1.26 (0.41–3.92)	0.6860		
OSA							
No	1608	1,140,454	1.41	Ref.			
Yes	2	844	2.37	1.68 (0.42–6.70)	0.4658		
Depression							
No	1570	1,123,495	1.40	Ref.		Ref.	
Yes	40	17,803	2.25	1.61 (1.17–2.20)	0.0031	1.21 (0.88–1.66)	0.2501
Anxiety							
No	1609	1,140,420	1.41	Ref.			
Yes	1	878	1.14	0.81 (0.11–5.73)	0.8295		
Headaches							
No	1437	1,036,655	1.39	Ref.		Ref.	
Yes	173	104,643	1.65	1.19 (1.02–1.39)	0.0300	0.92 (0.78–1.09)	0.3309
Liver diseases							
No	1339	987,052	1.36	Ref.		Ref.	
Yes	271	154,246	1.76	1.29 (1.13–1.47)	0.0001	1.12 (0.97–1.28)	0.1132
Peptic ulcers							
No	1120	893,273	1.25	Ref.		Ref.	
Yes	490	248,025	1.98	1.58 (1.42–1.75)	< 0.0001	1.13 (1.01–1.26)	0.0401
Cataract							
No	1260	1,070,736	1.18	Ref.		Ref.	
Yes	350	70,562	4.96	4.24 (3.76–4.77)	< 0.0001	1.90 (1.66–2.19)	<0.0001
CRVO							
No	1610	1,141,191	1.41	Ref.			
Yes	0	108	0.00	NA			
AMD							
No	1557	1,132,905	1.37	Ref.		Ref.	
Yes	53	8393	6.31	4.60 (3.50–6.05)	< 0.0001	2.11 (1.59–2.80)	<0.0001
Glaucomas associated drug							
No	787	649,874	1.21	Ref.		Ref.	
Yes	823	491,424	1.67	1.38 (1.25–1.52)	< 0.0001	1.10 (0.99–1.22)	0.0810

Rate, per 1000 person-years. OSA: obstructive sleep apnea; CRVO: central retinal vein occlusion; AMD: age-related macular degeneration. Adjusted HRs were measured by multivariate analysis using variables significant in measuring the crude HR.

**Table 3 ijerph-19-00305-t003:** Incidence and hazard ratio of glaucoma among patients with one or more components of metabolic disease.

Variables	*N*	Glaucoma No.	Person-Years	Rate	Adjusted HR (95% CI)	*p*
None	84,072	659	662,371	0.99	Ref.	
Only Hypertension	21,863	478	244,949	1.95	1.48 (1.31–1.67)	<0.0001
Only Diabetes	6365	163	68,391	2.38	2.14 (1.80–2.55)	<0.0001
Only Hyperlipidemia	11,490	239	138,752	1.72	1.79 (1.54–2.09)	<0.0001
Diabetes + Hypertension	439	17	4510	3.77	2.87 (1.78–4.66)	<0.0001
Diabetes + Hyperlipidemia	1094	35	12,979	2.70	2.63 (1.87–3.69)	<0.0001
Hypertension + Hyperlipidemia	680	12	8110	1.48	1.33 (0.75–2.36)	0.3275
Diabetes + Hypertension + Hyperlipidemia	105	7	1235	5.67	4.95 (2.35–10.4)	<0.0001

Rate, per 1000 person-years; Adjusted HRs were measured after controlling for age, gender, depression, peptic ulcers, cataract, and age-related macular degeneration.

**Table 4 ijerph-19-00305-t004:** Incidence rates of open angle glaucoma and Closure angle glaucoma and metabolism disease cohort to control cohort adjusted hazard ratios.

	Patients with One or More Metabolic Disease			Controls				
Outcome	Glaucoma (*n*)	Person-Years	Rate	Glaucoma (*n*)	Person-Years	Rate	Adjusted HR (95% CI)	*p*
Overall	951	478,927	1.99	659	662,371	0.99	1.66 (1.50–1.85)	<0.0001
Open angle glaucoma	487	478,927	1.02	292	662,371	0.44	2.03 (1.75–2.36)	<0.0001
Closure angle glaucoma	464	478,927	0.97	367	662,371	0.55	1.44 (1.25–1.66)	<0.0001

Rate, per 1000 person-years; Adjusted HRs were estimated after controlling for age, gender, depression, peptic ulcers, cataract, and age-related macular degeneration.

## Data Availability

We are not allowed to duplicate the database used in this study. Requests for data can be sent as a formal proposal to the NHIRD (http://nhird.nhri.org.tw, accessed on 30 July 2021).

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
