# Peer review of "Risk of Glaucoma Associated with Components of Metabolic Disease in Taiwan: A Nationwide Population-Based Study"

_ijerph, 2021, doi:10.3390/ijerph19010305_

Round 1
Reviewer 1 Report
Glaucoma is a well-known and much researched topic. However, longitudinal studies with large sample sizes such as your research article are very valuable in solidifying the knowledge base on the topic of glaucoma and metabolic syndrome. Do your research findings indicate any changes to current practices in screening patients with metabolic syndrome (especially diabetes) for glaucoma?
Reviewer 2 Report
English style and grammar need to be improved. All acronyms need to be extended when appear for the first time. Articles editing should be also improved. I think that tables and figures should be moved to the main section of results.
In general, paper seems to be interesting for readers and involving an important piece of data on glaucoma comorbidities based on huge sample sizes.
„For each patient with any component of MetS, two controls free of MetS were 103 randomly selected from the LHID2000, frequency-matched by sex, age (in 5-year bands) 104 and index year as controls of the non-MetS cohort (N = 84,072).” Could You explain this piece more? What LHID2000 means? What 5-years band means?
„OSA, depression, anxiety, 110 headaches, liver diseases, peptic ulcers, cataract, CRVO, and AMD.” Could You explain these acronyms in the paper?
Stat. analysis:
„The statistical significance 131 level was set as p < 0.05.” You should phrase it as „All analyses were performed with a significance level α = 0.05.” or similar. What variables exactly were analysed using t-test?
Results:
„with 40–64 years old.” What does it mean exactly?
„Figure 2 shows that the cumulative incidence of glaucoma was 1.5% greater in the 140 MetS cohort than in the controls (p <0.001).” in what period of time? 5 years?
„Compared with patients less 147 than 40 years old, the incidence of glaucoma was near 3-fold higher in those aged 40–64 148 years and 6-fold higher in the elderly.” So what was the age range of the examined groups? What about the mean and percentiles?
“WetS” a typo?
“Among patients having 158 two components of the MetS, diabetes mellitus was associated with a considerable in-159 creased risk of glaucoma.” It is a repeated of the previous information, what about risk related to other two components?
“glaucoma in women with MetS. 189
This study shows that the incidence of glaucoma is the highest for the elderly.” Please rephrase this sentence to not use „elderly”. Older adults or older patients seems to be more appropriate. Moreover, please be more specific: is the relationship between age and risk seems to be perfectly linear? Or maybe the risk is higher for certain ager groups? If so, please add results description in results section.
“Moreover, this study showed that some drugs could also induce glaucoma.” Please rephrase, You are rather describing correlation, not causation. So certain drugs are correlated with glaucoma. Please be more specific here: what class of drugs are You referring to here?
“ABCA1 mediates 231 lipid export and nascent HDL biogenesis [36], and caveolin 1 has been proven to be in-232 volved in lipid metabolism and its regulation [37].” Please remember about the proper way of editing of gene names in scientific articles
Table 1.
p-value in Hypothyroidism comes from chi-square test for independence? What is the conclusion then? You should describe it in the results section. Please remember, that high sample size means high statistical power in chi-square test. Did You hypothesis was that comorbidities values are related to MeTs occurrence?
Table 2 and 3
Adjusted HR Could You specify here in the table caption how the HR was adjusted? For what variables?
Table 4. what „comparison” means? Please add description in the table caption.
Figure 2. Kaplan-Meier method estimated cumulative incidence of glaucoma. Presents time of follow-up. Could You describe time of follow up more in the results? What was the mean time? What was the minimum? What was the maximum? How many patients were followed for 14 years? Please be as specific as possible and add this description to materials and methods or results section
Reviewer 3 Report
Title - location? population eg. adults or at risk or elderly?
Abstract - mention the population/location/database studied
Introduction
- There are inconsistencies in citation style eg. line 28, 36, 53... and I don't think these references are appearing in the reference list as well.
- Some details are not required eg. line 58-63. Keep the Introduction concise and focused on telling us about the research gap.
Methods
- Mets can be defined using established criteria eg. IDF, NCER ATP III, Harmonized and so on. Authors have to explain why such widely used criteria were not used in this instance. This is critical as it would affect the data analysis and results interpretation.
Results
- insert tables, figures within manuscript, not at the end and formatted as per journal's guideline.
- lack of important metabolic indicators eg BMI and WC has severely affected the interpretation.
Discussion and conclusion - no further comments as this depends on the results interpretation.
Round 2
Reviewer 2 Report
The above version has been successfully modified according to my comments. Well done!
Reviewer 3 Report
I appreciate the changes made by the authors to further improve the paper.
However, I still have reservation with the use "metabolic syndrome" to describe the clustering that has been made without using an established criteria as this is inaccurate. For example, FBG>5.6mmol/L will score 1 point if we use the Harmonized criteria, but this is not a diabetes diagnosis. So, by using diabetes as one of the criteria, we have missed out those whom we classify as prediabetes (5.6-6.9 mmol/L). I would suggest the authors to change all the terms and writing referring to metabolic syndrome to (perhaps) metabolic diseases throughout the document.
